# Impact of Oral Microbiota on Flavor Perception: From Food Processing to In-Mouth Metabolization

**DOI:** 10.3390/foods10092006

**Published:** 2021-08-26

**Authors:** Mathieu Schwartz, Francis Canon, Gilles Feron, Fabrice Neiers, Amparo Gamero

**Affiliations:** 1CSGA, Centre des Sciences du Gout et de l’Alimentation, UMR1324 INRAE, UMR6265 CNRS, Université de Bourgogne Franche-Comté, 21000 Dijon, France; francis.canon@inrae.fr (F.C.); gilles.feron@inrae.fr (G.F.); fabrice.neiers@u-bourgogne.fr (F.N.); 2Department Preventive Medicine and Public Health, Food Science, Toxicology and Forensic Medicine, Faculty of Pharmacy, University of Valencia, Burjassot, 46100 Valencia, Spain

**Keywords:** oral microbiota, flavor, perception, fermented beverages, wine, beer, glycosidases, carbon–sulfur lyases

## Abstract

Flavor perception during food intake is one of the main drivers of food acceptability and consumption. Recent studies have pointed to the oral microbiota as an important factor modulating flavor perception. This review introduces general characteristics of the oral microbiota, factors potentially influencing its composition, as well as known relationships between oral microbiota and chemosensory perception. We also review diverse evidenced mechanisms enabling the modulation of chemosensory perception by the microbiota. They include modulation of the chemosensory receptors activation by microbial metabolites but also modification of receptors expression. Specific enzymatic reactions catalyzed by oral microorganisms generate fragrant molecules from aroma precursors in the mouth. Interestingly, these reactions also occur during the processing of fermented beverages, such as wine and beer. In this context, two groups of aroma precursors are presented and discussed, namely, glycoside conjugates and cysteine conjugates, which can generate aroma compounds both in fermented beverages and in the mouth. The two entailed families of enzymes, i.e., glycosidases and carbon–sulfur lyases, appear to be promising targets to understand the complexity of flavor perception in the mouth as well as potential biotechnological tools for flavor enhancement or production of specific flavor compounds.

## 1. Introduction

The organoleptic perception experience during food intake is one of the main drivers of food acceptability and consumption. Flavor perception plays a major role in this organoleptic experience. It is a multimodal perception corresponding to functional integration of information from the chemical senses: olfaction, gustation, and nasal and oral somatosensory inputs. By providing information on the chemical composition of food, flavor allows the organism to evaluate food quality. However, the nature of the chemicals that reach chemoreceptors can be impacted by the perireceptor environment [1,2]. Recent investigations have revealed a role of enzymatic degradation of flavor compounds on flavor molecules and perception [2]. Thus, salivary disorders that appear with age can impact food enjoyment and intake, leading to malnutrition [3]. In addition to human metabolism, several recent studies have explored the influence of the oral microbiota on taste perception and food choices [4,5,6]. With regard to food aroma, some specific microorganisms of the oral flora are able to generate fragrant molecules in the mouth, thus participating in the perception of food flavors [7,8]. These results suggest that variations in microbial composition in the mouth could be a possible cause of differences in perception. While these concepts seem particularly innovative in food science focused on the consumer, the molecular mechanisms involved are actually not new from the perspective of food microbiology. Indeed, microorganisms have been used for thousands of years for the production and fermentation of food products, such as bread, wine, and beer. Fermentation leads to the production of alcohol, which preserves beverages from microbiological contamination, as well as carbon dioxide, which affects the volume and texture of bread dough. It also produces odorant molecules via specific metabolic pathways [9]. For instance, pyruvic acid is generated by glycolysis and can be metabolized to higher alcohols, short-chain fatty acids, and carbonyl compounds during the fermentation of bread doughs [10]. These compounds play an important role in the flavor of the final product. Some of these metabolic pathways are similar to those found in some oral microbes, potentially producing similar flavor compounds. This is the case for the metabolization of glycosides and cysteine conjugates by the action of certain microbial enzymes, increasing volatiles and flavor perception [8,11,12,13,14]. Thus, the metabolic activity of oral microorganisms is likely to affect both the chemical nature of flavor compounds and flavor perception and in fine consumption.

In this context, the objectives of this review are (i) to present the state of knowledge on the links between oral microbiota and flavor perception in foods and (ii) to identify and discuss common metabolic pathways between oral microorganisms and strains involved in food processing. For this purpose, we first introduce the current knowledge on oral microbiota. Second, recent studies showing links between oral microbiota composition and chemosensory perception are presented. Finally, the most important metabolic pathways involved in the production of food products are discussed to establish a parallel with reactions catalyzed by microorganisms in the mouth and generating flavor molecules. This work aims to highlight important reaction pathways in the mouth that are crucial for food choices but are also of interest for the production of food products, such as fermented beverages (e.g., wine and beer).

## 2. Characteristics of the Oral Microbiota

The oral cavity is a niche that hosts more than 700 species of bacteria and other microorganisms (including fungi, parasites, and viruses), constituting the oral microbiota [15]. Colonized sites include the tongue and its dorsum [16], the tissues of the oral mucosa, soft and hard palate, and teeth. The oral microbiota is essentially made of facultative anaerobes, such as *Streptococcus* and *Actinomyces* species, and strict anaerobes, such as *Bacteroidaceae* and *Fusobacteriaceae* in the sites reduced in oxygen (e.g., subgingival area) [17]. On the surface of teeth, microorganisms from multiple species form biofilms, which promote interactions between species [18]. Oral microorganisms are not exclusively bacteria. Several fungal genera, such as *Candida, Cladosporium*, *and Saccharomyces,* constitute the oral mycobiome, which is still poorly studied [19]. In the oral cavity, bacteria and fungi interact [20]. The oral microbiome differs between healthy individuals and changes drastically upon eating and tooth brushing [21], although studies have also indicated a relative stability of the oral microbiota [22].

### 2.1. Development of the Microbiota throughout Age

The oral microbiota varies throughout life, and its development is influenced by external factors [23]. During the first days of an infant’s life, the bacterial species present are primarily *Streptococcus*, *Veillonella*, and *Lactobacillus* [24]. Biofilm development is then limited due to the absence of teeth, which provide a hard surface necessary for adhesion. Gradually, during the first years of the child’s life, the microbiota expands with the appearance of additional species, such as *Gemella*, *Actinomyces*, and *Neisseria*. Interestingly, breastfeeding habits were found to significantly decrease some bacteria, such as *Actinomyces* and *Porphyromonas,* several months later [24] but increase *Streptococcus* [25]. Additionally, it was shown that antibiotics administered during the first years of life continue to impact the development of oral microbiota several years later [24].

### 2.2. Consequences of Oral Pathologies on the Microbiota

Most oral pathologic situations occur when bacterial homeostasis is imbalanced, leading to the accumulation of pathogenic biofilms (also known as dental plaques) [18]. This phenomenon is at the origin of common oral diseases, such as gingivitis (plaque around the gingival margin) or periodontitis (inflammation of the periodontal tissues). Dental caries are induced by a high-sugar diet, leading to an abnormal increase in species such as *Streptococcus mutans*, which produces acidic compounds that damage teeth [17]. Halitosis, commonly known as bad breath or oral malodor, results from the accumulation, on the tongue surface, of volatile sulfur compound-producing bacteria, such as *Porphyromonas* and *Fusobacterium* [26]. Oral microbiota dysbiosis also occurs during the development of oral carcinoma [27].

In addition to the oral diseases mentioned above, the oral microbiota is involved in several systemic diseases. For example, alterations of the oral microbiota occur in individuals with diabetes. It has been suggested that an increase in blood glucose concentration in diabetic patients disrupts the homeostasis of the oral microbiota [28]. In obese patients, a significant decrease in microbial diversity has been observed in comparison with healthy individuals [29], while this difference was significantly less noticeable at the level of the gustatory papillae [5]. In xerostomia (dry mouth feeling) caused by Sjogren’s syndrome (altered salivary glands), an increase in infections caused by nonoral bacteria has been shown [30]. As saliva plays a key role in the maintenance of oral heath, a decreased salivary flow, and therefore reduced immune protection by saliva, likely explains this disorder [28].

### 2.3. Saliva Microbiota

Saliva is the fluid secreted by the salivary glands, providing essential protective functions for health [31]. Saliva has a maintenance role toward microorganisms colonizing the oral cavity due to the action of different proteins: lysozyme, immunoglobulins, lactoferrin, and the peroxidase system [32]. This biological fluid also promotes lubrication of the oral cavity and has a role in gustation and the release of flavor molecules in the mouth [1,33]. The microbiota of saliva is similar to the microbiota of the oral mucosa and tongue [34]. In this way, saliva is often used to study the oral microbiome because it is easily collected. Furthermore, saliva has a beneficial effect on the flora because it solubilizes and transports the nutrients ingested during food intake to the different colonization sites in the mouth. It contains salivary proteins that can serve as nutrients for certain anaerobic proteolytic species, such as *Porphyromonas* and *Prevotella* [34]. More than 2000 bacterial proteins from 50 bacterial genera have been identified in saliva [21].

### 2.4. Influence of External Factors and Diet

The microbiota is shaped more by the environment than the genetics of the host, while salivary microbiome composition established during familial upbringing can persist over a timescale of years [35]. The impact of many external factors on the oral microbiota has been studied [28]. To cite a few, modulation of the oral microbiota has been demonstrated following physical exercise (increase of nitrate-reducing activity by oral microbiota) [34], exposure to cigarette smoke (elevated levels of *Streptococcus*, *Prevotella*, and *Veillonella* among smokers) [36] or altitude (increase of *Prevotella* and decrease of *Streptococcus* in high-altitude Tibetans) [37]. While the impact of diet on the gut flora microbiota is significant and very well studied, the impact of diet on the oral microbiota was suggested to be moderate according to limited studies [28,38]. In reality, the identification of such correlations is not trivial and may require the development of specific methodologies [39,40]. A few studies have highlighted the prevention of periodontitis caused by the proliferation of pathogenic anaerobes by specific diets [41]. Tea-rich diets have been shown to increase oral microbial diversity and increase the abundance of the genera *Fusobacteriales* and *Clostridiales* [42]. In contrast, oolong tea consumption has been shown to decrease oral microbial diversity and reduce species, such as *Streptococcus* sp., *Prevotella nanceiensis*, and *Fusobacterium periodonticum* [43]. Tea contains flavan-3-ol compounds, such as epigallocatechin gallate, that have antimicrobial effects [44]. A coffee-rich diet has been shown to cause an increase in the abundance of the genus *Granulicatella* [42]. Some bioactive dietary compounds have been identified as exerting a reducing effect on oral pathogenic bacteria populations, such as theaflavin supplemented in toothpaste [45]. One study suggested the existence of an association between sugar intake and oral microbiota ecology and a response of oral microbiota to sugar beyond acidogenic species [46]. Such a correlation needs to be further confirmed in the future.

## 3. Oral Microbes Modulate Chemosensory Perception

Chemosensory perception results from the activation of chemoreceptors by a large diversity of compounds belonging to different chemical families. Recent research indicates that the metabolic activity in mouth is susceptible to modify both the quality and the quantity of the compounds activating the receptors [2,47]. Despite the presence of a high microbe diversity in the oral cavity, the knowledge of the contribution of the oral microbiota on the salivary metabolome is just emerging. Up to now, only few studies have focused on the link between the salivary metabolome and the salivary microbiome. Three mechanisms allowing the oral microbiota to modulate the host chemosensory perception can be highlighted. First, the generation of metabolites by microbial enzymes can activate or modulate the activation of the host chemoreceptors [7]. Second, the bacterial metabolization of exogenous molecules participate to the termination of their perception [48]. Third, the microbiota can manipulate the chemical senses of the host by changing the receptors density [49,50].

### 3.1. Modulation of the Host Taste and Smell Perception

Salivary metabolites can either be produced by the oral microbiota or by host enzymatic activity [1,2]. Metabolites can modulate taste and smell perception at two levels. The first concerns the basal-level production of flavor-active compounds, which influence the threshold of perception of these specific molecules, as well as the metabolization of food compounds into metabolites that can activate taste and smell receptors. Furthermore, the metabolization of taste and smell molecules into new molecules without chemosensory properties also contributes to modulation of the chemical senses by decreasing the quantity of flavor compounds.

Short-chain fatty acids, acetate and propionate, are the most abundant salivary metabolites [51]. Acetate and propionate have to be generated by bacterial activity in the mouth because they are not present in the parotid saliva, and because there is a strong correlation of bacterial load with the concentration of these molecules. They are generated from endogenous compounds, mainly issued from the saliva secreted by the host, and from exogenous nutrients coming from food. The basal concentrations of taste compounds, such as salt, in saliva influence their perception threshold through adaptation [52,53]. More recently, it was suggested that the same adaptative mechanism could impact the threshold of fat perception via endogenous production of fatty acids [54]. Microbiota could play a role at the level of this threshold, as it was also proposed that the lipolysis of fatty acids in saliva is driven in part by microbial lipase [55,56]. Indeed, genes encoding secreted lipases are not expressed in human lingual tissue [57], supporting an alternative origin of salivary lipolytic activity. Additionally, the orosensory detection of lipids was shown to be directly linked to the existence of specific microbiota in saliva and independent of BMI status [5].

Amino acids are perceived as umami, especially glutamate, which is one of the most abundant amino acids in body fluids. Contrary to salt or fatty acids, the basal concentration of glutamate in saliva does not modify its threshold of perception [58]. However, the salivary glutamate concentration may influence perceived pleasantness [58]. Unpleasantness ratings of concentrated solutions of monosodium glutamate are higher in subjects with low salivary glutamate concentrations than in subjects with high salivary glutamate concentrations [58]. Many mouth bacteria, such as *P. gingivalis* or *Fusobacterium* species, utilize saliva glutamate in diverse reactions (deamination, decarboxylation) [59], consequently modulating its basal concentration. Thus, we hypothesize that bacteria contribute to the pleasantness of monosodium glutamate perception.

Concerning the physiology of olfactory perception, it was shown that microbial enzymes can impact it as well. François et al. [48] reported that the amplitude of odorant responses was increased in germ-free mice. The same mice showed altered kinetics of the olfactory response in association with a decrease in the concentration of olfactory xenobiotic-metabolizing enzymes [48]. A decrease in the metabolic activity of these enzymes can impact the odorant concentration in the perireceptor environment and, thus, the kinetics and amplitude of the olfactory response due to a higher adaptation. At the same time, transcription of genes encoding olfactory receptors is also decreased, which could affect the intensity of the sensory response.

Microbial enzymes could also play a direct role in perception by generating new odorants in the respiratory and olfactory epithelium. For instance, the oral mucosa was reported to metabolize aroma compounds into new odorant compounds [2,60]. Moreover, some odorants were reported to be specifically produced by bacterial enzymes in the mouth during food oral processing. For instance, glycoside-derived aroma compounds are mainly produced by bacterial enzymes because the majority of glycosidase enzymes in the mouth are produced by bacteria [61,62]. The oral microbiome contributes to the interindividual variation in saliva hydrolysis capacity, driving aroma compound formation [7].

It appears that the impact of taste compounds metabolism on their perception has to be more deeply explored, first in relationship with host enzymes localized in the mouth (as it was studied in case of odorant metabolization in the olfactory and respiratory epithelium [63,64]), and second, in relationship with microbial enzymes. Importantly, the microbiota present in the close vicinity of gustatory papillae has been poorly investigated but could play a crucial role by generating metabolites directly activating or modulating the host taste receptors. In addition, the oral microbiota could also influence in-mouth molecular mechanisms by modulating environmental features, such as pH and redox status, indirectly impacting perception.

### 3.2. Modulation of the Host Receptors Expression of Host Genes Encoding Receptors 

Different studies support the fact that microbes can regulate the expression of taste receptors genes. Germ-free mice present a decrease in intestinal satiety peptides associated with an increase in oral nutrient detection [65]. This increase could be due to the increase in the lingual CD36 receptor mRNA [65], as this receptor has been proposed to be involved in fatty acid detection [66].

One study showed that children presenting lower sensitivity to the sweet taste had more *S. mutans* isolated from mouth washes [67]. At the same time, it is known that *S. mutans* biofilm formation is favored by higher sucrose consumption. Thus, a high sugar diet can modify the equilibrium of the different bacterial communities and, as a result, the host inflammatory response. Indeed, bacterial lipopolysaccharides, which are produced by certain bacteria, lead to the production of cytokines that drive a host inflammatory response. This inflammatory response consequently decreases the number of taste receptor cells in mice [49,50].

Interesting perspectives are open in these emerging fields regarding the numerous additional mechanisms observed for host behavior manipulation by gut microbiota. These additional mechanisms can include the induction of dysphoria and hormone level modulation, as well as hijacking the host’s nervous system [68].

## 4. Common Pathways for the Microbial Production of Flavor Compounds in Fermented Products and in the Mouth by Oral Microbiota

Flavor is one of the most relevant attributes determining food quality and acceptance by consumers. Odor and taste in fermented products, such as wine and beer, are determined to a large extent by the action of microorganisms either involved in the fermentation process or present in the oral cavity. One of the most noteworthy mechanisms of aroma generation is the metabolization of precursors through the action of microbial enzymes. This occurs in the product as well as the oral cavity during chewing/drinking and can be carried out by different enzymes, such as glycosidases or carbon–sulfur lyases (Figure 1) [11,12,13]. In the following lines, we present two main categories of such precursors, namely glycoside conjugates and cysteine conjugates.

### 4.1. Glycoside Conjugates

#### 4.1.1. Glycosides as Aroma Precursors

One of the most abundant nonvolatile aroma precursors in plant-based food products are glycosides. These compounds can release odorants through hydrolysis [7]. The sugar part of the glycosides can be just glucose (β-d-glucopyranosides) or glucose conjugated with a second sugar unit of α-l-arabinofuranose, α-l-rhamnopyranose, β-d-xylopyranose, or β-d-apiofuranose [14]. In this way, the sugar component of glycosides can consist of many different types of mono- and disaccharides, and the non-sugar part (aglycone) can include a wide range of aromas [7,12]. The hydrolysis of monoglucosides only requires the action of a β-glucosidase, whereas the hydrolysis of disaccharide glycosides requires the sequential action of two enzymes, a proper exoglycosidase (α-l-arabinosidase, α-l-rhamnosidase, β-d-xylosidase, or β-d-apiosidase) to remove the outermost sugar molecule and a β-glucosidase to remove the remaining glucose [14,69]. 

#### 4.1.2. Glycosides in Wine

In wines, the potential of glycosidic nonvolatile aroma precursors to improve global aroma is very high due to aglycones, which are generally potent flavor compounds with low sensory thresholds and appealing sensory properties [12,70]. In addition, these precursors appear in much greater quantities than free aroma compounds, up to 10-fold [12], due to their higher affinity for the aqueous phase due to the hydrophilic property of the sugar part of the glycosidic precursors. The percentage and type of glycosidic precursors vary among different grape varieties. The main known aglycones are terpenes, C_13_-norisoprenoids, volatile phenols, C_6_ compounds, aliphatic alcohols, aliphatic acids, benzenic compounds, and phenolic acid derivatives (Table 1), which can provide floral, fruity, or toasted notes, among others [12,70,71].

In general, *Saccharomyces cerevisiae*, the most common yeast added to wines as a starter, does not perform remarkably in this sense [14]; however, several *Saccharomyces* hybrids and non-*Saccharomyces* yeasts have shown an enormous potential to increase varietal aroma through the action of different glycosidases, such as β-glucosidases (Table 1). Examples of these yeasts are hybrids among *S. cerevisiae*, *S. uvarum*, and *S. kudriavzevii* as well as non-*Saccharomyces* yeasts, such as *Pichia anomala, Candida molischiana*, *C. wickerhamii, Hanseniaspora uvarum*, and *Metchsnikowia pulcherrima* [12,72,73,74]. In addition, other yeast genera have been demonstrated to present β-glucosidase activity, such as *Debaryomyces*, *Kluyveromyces*, *Saccharomycodes*, *Schizosaccharomyces*, and *Zygosaccharomyces*. However, the in vitro substrates used for β-glucosidase detection are hydrolyzed by glucanases as well; therefore, it must be taken into account that these two activities can be confounded [75]. In fact, *Saccharomyces* presents different exo-1,3-β-glucanases, encoded by the genes *EXG1* and its paralogs *SPR1* and *EXG2,* and the activity of these enzymes has been related to glycoside hydrolysis [72]. In addition, according to the *Saccharomyces* Genome Database, the *EGH1* gene encodes a β-glucosidase with a broad specificity for aglycones, which has been related to hydrolysis of flavonoid glucosides [76] and could have a role in aroma improvement.

Microorganisms responsible for malolactic fermentation, especially the species *Oenococcus oenii,* have also the potential to improve wine aroma [14] (Table 1). This type of fermentation is characteristic of red wines originating in cold regions, where the acidity of grapes is higher. Several potential genes encoding glycosidases have been identified in *Oenococcus* and in other lactic acid bacteria (LAB), such as *Lactobacillus* and *Pediococcus*, as well as a gene encoding a β-glucosidase [14]. Ethanol, residual sugars, temperature, pH and LAB strain have been demonstrated to be important factors levelling enzymatic activity. Nevertheless, more investigations are needed to properly elucidate the relationship between gene expression and β-glucosidase activity [14].

#### 4.1.3. Glycosides in Beer

Odorless glycosides can also appear in beer. In this case, the main sources of these aroma precursors are hops and wood barrels used for maturation [13,69]. These glycosides have the potential to increase or modify the aroma of hops, since several key compounds can be released, such as linalool (citrus, floral, and aniseed flavors), methyl salicylate (wintergreen, mint, and spice flavors), or raspberry ketone (raspberry aroma) [13,77]. The concentration of hop glycosides seems to be predominantly cultivar-dependent [77] and their hydrolysis mainly occurs in beers produced through spontaneous fermentation and fruit maceration. Examples of these types of brewing processes are Lambic and Gueuze beers, which employ cherries (Kriek), or sour beer varieties [78]. This hydrolysis can occur through two mechanisms: acidic hydrolysis due to the low pH of this type of beer or enzymatic hydrolysis either by added enzymes or yeast enzymes [69]. In the latter case, the yeast genus responsible is essentially *Brettanomyces/Dekkera*, mainly the species *B. bruxellensis, B. custersii*, and *B. anomalus* (Table 1), which are able to synthesize β-glucosidases [13,78]. This mechanism of increasing beer aroma is still poorly explored [13] but opens an interesting route for beer aroma improvement through bioflavoring. 

Brewing *Saccharomyces* strains do not have 1,4-β-d-glucosidase (BGL) activity, and only strain-dependent exo-1,3-β-glucanase (EXG), mainly encoded by the *EXG1* gene, presents low to moderate activity [78]. *EXG1* gene shows high expression during yeast exponential growth but is repressed during fermentation due to hypoxic conditions. Nevertheless, *Saccharomyces* strains presenting high EXG activity showed BGL-like activity against hop glycosides in brewing but lower activity than *Brettanomyces* strains [78]. For instance, *B. custersii* LD72 showed higher release of sour cherry glycosides compared with *Saccharomyces* ale strains in refermentation experiments, either with the addition of amygdalin (a cyanogenic glycoside present in the seeds of the sour cherry) or including whole cherries, cherry pulp, cherry juice, and cherry stones [69].

#### 4.1.4. Glycosides in the Oral Cavity

The presence of microbial glycosidase enzymes in saliva was demonstrated as early as in 1954 [79]. In 1999, the in-mouth hydrolysis of glucosides was observed and linked to flavor perception [80]. More recently, the β-glucosidase activity of certain bacteria of the oral microbiota was described. This is the case for the genera *Streptococcus* and *Prevotella* [7]. However, limited research has been carried out to elucidate the role of oral microbiota in the release of volatiles from glycosides, and most of these studies only tested the in vitro potential without taking into account the tasting time frame or the conditions of the mouth environment [7]. In fact, the hydrolysis of glycosides occurs relatively slowly during wine processing and storage, whereas this process must occur very quickly in the oral cavity while drinking the beverage [7]. The flavor of the aglycone is perceived by retronasal olfaction after the action of glycosidase enzymes within seconds of placing the glycoside in the mouth. This phenomenon has been demonstrated for glycosylated monoterpenes and phenols as well as for other compounds, such as hexyl glucoside, despite being highly variable between individuals [7]. In the case of taste sensitivity, a positive correlation with some bacterial phyla was reported. The existence of *Actinobacteria* and *Bacteroidetes* at the tongue surface increases the sensitivity to bitterness [4]. The same mechanism could exist for retronasal olfaction, which can be affected by certain microbial species.

Despite the importance of oral microbiota in the hydrolysis of glucosides, the main factor limiting retronasal perception of released aroma compounds is the odorant threshold [7]. This fact was proven in a study carried out by Parker et al., including guaiacyl glucoside, geranyl glucoside, and other glycosides extracted from the Gewürztraminer grape variety [7]. This parameter would explain the interindividual perception differences that classify people into “tasters” and “nontasters”.

#### 4.1.5. Metabolization of Glycosides in the Oral Cavity during Alcoholic Beverage Consumption

Studies dealing with the hydrolysis of glycosides in the oral cavity are limited, as previously commented. Some studies carried out after wine consumption showed that the aroma compounds released from glycosides can enhance the complexity, intensity, and persistence of aroma during the consumption of this beverage [11,70,81].

Most volatile phenols, such as guaiacol, syringol, or *m*-cresol, appear in wines as disaccharide glycosides in relatively high concentrations, especially in grapes that have undergone smoke exposure. In fact, some of these volatile phenols have been identified as significant contributors to smoky aroma and taste in wines [81]. The hydrolysis of these glycosides was observed by using in vitro and in vivo approaches in smoke-affected wines even under the low pH and high ethanol conditions typical of this beverage [81] (Table 1).

Muñoz-González et al. exposed wine glycosides isolated from white grapes to fresh saliva and, as a result, released different types of odorant molecules, such as terpenes, benzenic compounds, and lipid derivatives [82] (Table 1). This activity was attributed to the oral microbiota, because aroma compounds were not released when incubation was carried out with saliva enzymes but without oral microorganisms.

The hydrolysis of glycosides seems to be bacteria-dependent and subject to large interindividual variability [70,81]. This variability seems to be linked rather to qualitative than to quantitative differences in the microbiota composition. This variability might also be linked to other human physiological parameters, such as saliva composition, oral mucosa temperature or air volume changes in the oral cavity [70]. Despite the relatively short residence time of wine in the mouth, recent investigations point out the possible interactions between nonvolatile compounds from wine and oral and pharyngeal mucosa. Indeed, aroma compounds can interact with the thin layer of proteins at the surface of the oral mucosa, called the mucosal pellicle [60], which determines the surface properties of the oral mucosa [83]. These interactions may increase the time available for hydrolysis of glycosides [82]. For instance, a reduction in aroma release has been observed in red wines compared with white and synthetic wines when these products were exposed to human saliva. This reduction in aroma release was observed in most of the assayed aroma compounds independently of their chemical structures [82]. Regarding beer, no studies in this sense have yet been carried out, but aroma release and modifications by oral microbiota can be expected.

### 4.2. Cysteine Conjugates

#### 4.2.1. Cysteine Conjugates as Aroma Precursors

Volatile sulfur compounds are generally present in small amounts in foods. However, their contribution to the overall flavor of food is strong due to their particularly low perception thresholds [84]. They significantly participate in the typical flavor of fermented products, such as wine, beer, cheese [85], and some fruits [86]. Cysteine conjugates constitute an important class of sulfur aroma precursors. These compounds are found in plants and, therefore, in a number of plant-based foods including onions, bell pepper, some fruits, wine, and beer [8,87,88]. These molecules consist of a cysteine group linked by a carbon–sulfur bond to an organic group. Cysteine conjugates have little or no odorant properties due to their low volatility. Metabolization of these compounds by the action of microbial enzymes called carbon–sulfur lyases (C–S lyase) leads to the formation of molecules bearing a free thiol function and having odorant properties [89,90].

#### 4.2.2. Cysteine Conjugates Metabolism and Formation of Sulfur Aroma in Fermented Beverages

In the case of fermented beverages, the presence of specific sulfur compounds comes from the metabolism of cysteine conjugates by the microorganisms used during fermentation. In 1998, some of these compounds were identified in the Sauvignon white wine by Tominaga et al.: 4-mercapto-4-methylpentan-2-one, 4-mercapto-4- methylpentan-2-ol and 3-mercaptohexan-1-ol (Table 2) [91]. These compounds are derived from the hydrolysis of nonvolatile cysteinyl precursors present in grape must after metabolization by yeast. Thiols are generated by the action of C–S lyases catalyzing the dissociation of C–S bonds. The presence of precursors has been reported in several grape varieties, such as Semillon, Chardonnay, and Riesling [92]. The corresponding flavor compounds are of relevance because they are related to the enhancement of several aroma parameters during wine consumption, such as complexity, intensity, and persistence [11]. Regarding their origin, it has been suggested that some of these compounds, such as S-3-(hexan-1-ol)-l-cysteine, are generated in grapevine by the catabolism of glutathione precursors, such as S-3-(hexan-1-ol)-glutathione [93]. These molecules are probably synthesized in grapevine by detoxification systems, such as glutathione transferases [87].

The presence of polyfunctional thiols generated from cysteine-conjugated precursors, such as 3-methyl-2-buten-1-thiol, 2-mercapto-3-methylbutanol, and 3-mercapto-3-methylbutanol, has also been reported in Lager beer (Table 2) [94]. These precursors are found in several hop varieties [88] and are likely synthesized by enzymes of the glutathione transferase family [95], similar to the one in grapevine. The addition of a commercial C–S lyase (tryptophanase) to a hop solution has been shown to trigger the formation of some free thiols [88]. Their formation in beer is triggered during alcoholic fermentation by the enzymatic action of C–S lyases from *S. cerevisiae* or other yeasts. Notably, a strain of *Pichia kluyveri* has been patented to improve thiol levels during beer fermentation [69]. Interestingly, Belda et al. reported the identification of several *Saccharomyces* strains as well as non-*Saccharomyces* yeast strains, such as *Torulaspora delbrueckii*, *Meyerozyma guilliermondi*, and *Kluyveromyces marxianus,* capable of enhanced thiol release through increased lyase activity [96].

The yeast genes responsible for C–S lyase activity are *IRC7* [89] and *STR3* [97]. These genes encode cystathionine β-lyases, which are pyridoxal-5′-phosphate-dependent enzymes catalyzing the dissociation of the C–S bond of various substrates, such as l-cystathionine, l-cysteine, and l-cystine, as well as precursors of thiol aroma compounds, such as 4-sulfanyl-4-methylpentan-2-one and 3-sulfanylhexanol [69]. Engineering strains for overexpression of these genes is a way to improve the release of varietal thiols to enhance wine flavor [89,97]. Furthermore, the presence of C–S lyases in lactic bacteria, e.g., *Lactobacillus* species [90], suggests potential applications for improving the flavor of certain red wines undergoing malolactic fermentation and for other food products.

#### 4.2.3. Metabolization of Cysteine Conjugates in the Oral Cavity

As mentioned above, cysteine conjugates are also found in a number of nonfermented foods in precursor forms. This is the case, for example, for onions, peppers, garlic, and certain exotic fruits [8]. In addition, cysteine conjugates can also be generated in heated foods through Maillard reactions. The formation of S-furfuryl-l-cysteine and S-(2-methyl-3-furyl)-l-cysteine from xylose and cysteine heated at 100 °C for 2 hours has been demonstrated [98].

These non-odorant precursor compounds generate flavor compounds in the mouth through the action of certain microorganisms in the oral microbiota [8,98] following enzymatic mechanisms similar to those employed by fermentative microorganisms (Table 2). In a pioneering study, Starkenmann et al. showed the oral metabolism of cysteine conjugates, such as S-(R/S)-3-(1-hexanol)-l-cysteine, S-(1-propyl)- l-cysteine and S-((R/S)-2-heptyl)-l-cysteine by sensory, microbial, and molecular approaches. In this study, it was shown that the salivary anaerobe *Fusobacterium* metabolizes such compounds to their corresponding thiols [8]. It is imperative to keep in mind that *Fusobacterium* spp. has an extensive enzymatic arsenal, able to metabolize various sulfur compounds, including cysteine conjugates [99]. Later, the influence of saliva on the metabolization of sulfur compounds from raw cabbage extracts was shown in relation to their perception [100]. In the same study, the degradation of S-methyl-l-cysteine sulfoxide present in cabbage under the action of microbial C–S lyases into various flavor compounds was suggested. Starkenmann et al. [8] showed that free thiols generated in the mouth are detected within seconds to minutes, pointing out the relevance of these mechanisms in flavor perception. Furthermore, saliva has been shown to play an additional role in trapping free thiols, probably through the action of salivary proteins [8]. Thiols can also be oxidized by salivary oxidant compounds, such as hypothiocyanite ions [33].

## 5. Conclusions and Perspectives

Food flavor is a key attribute that determines quality and acceptance by consumers. However, flavor perception is different among individuals, and part of this dissimilar sensitivity could be explained by the oral microbiota. Evidence suggests that the oral microbiota could play an important role in taste modulation. In this context, one challenge for the future will be to investigate the composition of oral microbiota around gustatory papillae. Qualitative differences in the microbial species present in the mouth may lead to different metabolizations of aroma compounds and their precursors, thus leading to different retronasal olfactive responses. Limited studies have been carried out in this sense, and further approaches, especially in vivo, therefore need to be performed to understand the complex mechanism of flavor perception and its interindividual awareness. Profile knowledge of the metabolic pathways related to aroma synthesis and release from precursors of fermentative microorganisms could aid in unravelling the homologous biochemical pathways of microorganisms present in the mouth. Alternatively, knowledge gained on oral microbial enzymes will also be useful for the design of molecular tools optimized for aroma compound production or enhancing the flavor intensity of specific food products. In this review, two families of enzymes have been highlighted as potential targets for future studies. Glycosidases and C–S lyases are such enzymes that produce flavor compounds in the mouth and in fermented beverages. While these mechanisms are known, evidence of the entailed genes remains scarce. Enzymes from the oral microbiota are very poorly studied and should be an upcoming area of research, aided both by genomic and proteomic data exploration as well as modern techniques of biochemistry and molecular biology. These results will be of relevance for both flavor perception understanding and fermented food flavor enhancement.

## Figures and Tables

**Figure 1 foods-10-02006-f001:**
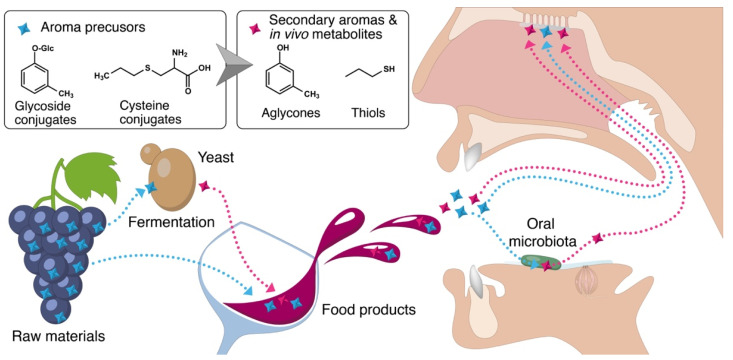
Pathways enabling the formation of flavor compounds from aroma precursors in alcoholic beverages and the mouth.

**Table 1 foods-10-02006-t001:** Production of flavor compounds from glycoside precursors by microorganisms in fermented beverages and in the oral cavity.

Localization	Flavor Compounds	Producing Microbial Species	Food Product
Fermented beverages	Terpenes C_13_-norisoprenoids Volatile phenols C_6_ compounds Aliphatic alcohols Aliphatic acids Benzenic compounds Phenolic acid derivatives	*Saccharomyces* hybrids, *Pichia anomala,* *Candida molischiana,* *Candida wickerhamii, Hanseniaspora uvarum, Metschnikowia* *pulcherrima*	Wine [12,70,71]
	Terpenes C_13_-norisoprenoids Benzenic compounds	*Oenococcus oenii*	Red wine [14]
	Terpenes Aliphatic alcohols	*Brettanomyces* *bruxellensis, B. custersii, B. anomalus*	Beer, special fruit beers [13,78]
Oral cavity	Volatile phenols	unknown	Smoke affected wines [81]
	Terpenes Benzenic compounds Lipid derivatives	unknown	White grapes [82]

**Table 2 foods-10-02006-t002:** Production of flavor compounds from cysteine precursors by microorganisms in fermented beverages and in the oral cavity.

Localization	Flavor Compound	Producing Microbial Species	Food Product
Fermented beverages	4-mercapto-4-methylpentan-2-one	*S. cerevisiae, E. limosum*	Sauvignon wine [91,92]
	4-mercapto-4-methylpentan-2-ol		
	3-mercaptohexan-1-ol		
	3-methyl-2-buten-1-thiol	*S. cerevisiae, P. kluyveri*	Lager beer [69,94]
	2-mercapto-3-methylbutanol		
	3-mercapto-3-methylbutanol		
Oral cavity	(R/S)-3-sulfanylhexan-1-ol	*F. nucleatum*	Grapes [8]
	1-propanethiol		Onion [8]
	2-heptanethiol		Bell pepper [8]

## Data Availability

Not applicable.

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
