# Peer review of "Impact of Oral Microbiota on Flavor Perception: From Food Processing to In-Mouth Metabolization"

_foods, 2021, doi:10.3390/foods10092006_

Round 1

Reviewer 1 Report

Investigation of the role and effects of oral microbiota in organoleptic and flavour perception is an interesting area of researches. Enzymatic reactions occurred by the oral microbiota have effect on aroma compounds, and therefore flavour perception that can influence the consumer purchase/buyer decision process, respectively. As author concluded, it can be possible to use ’biotechnological tools’ for flavor enhancement.

The review manuscript is well structured. The relevancy of the study and research motivations are well defined. Section 2 summarizes well the change of oral microbiota as a function of age, oral pathologies, diet and external factors. Section 3 is a good summary on modulation of chemosensory perception by oral microbiota. Section 4 is the mre detailed an maybe the most interesting part of the study. Establishments are vaulabe not just for the science but also for practice. Establishments and information based on relevant references. The review manuscript is generally well written and it contains interesting results for the readers.

Comments.

I suggest the authors to give the DOI.s for references (if available).

I suggest the authors to consider to give the information in section 4.1.2-4.1.4 and 4.2.2 in tables, as well.

Reviewer 2 Report

The authors need to follow the following instructions to improve this manuscript.

  • The authors should limit to use ‘we’ words. For instance, Line 12, 17 and followed the entire manuscript.
  • Please use more keywords that are not in the title of this manuscript
  • Add more information in the abstract based on the best findings of this manuscript.
  • Line 55 (in fine): Why use Italic?
  • Line 168-170: Please add reference if possible
  • Line 209-210: François and collaborators reported that the amplitude of responses to odorants is increased in germfree mice [63]. The authors may write as: François et al. [63] reported that the amplitude of odorant responses is increased in germfree mice.
  • Line 253 (his): What is the meaning of his?
  • Line 307 (In addition, and according to Saccharomyces): Check this sentence
  • Line 323 (On the other hand,): Why started with “On the other hand” in the new para?
  • Line 369-372: Very large sentence. Please rewrite this sentence.
  • Line 389 (Muñoz-González and collaborators): The authors may write Muñoz-González et al. [serial number]. Follow the entire manuscript (Line 452-455, Line 476, Line 486-487, ---). Please check the previously published papers of FOODS.
  • Line 389-393: Rewrite these sentences.
  • Line 409-410: Please merge with line 408.
  • Line 431: Check the gap
  • Line 436: Variety/cultivar usually writes in the inverted comma. Please use the first time inverted comma and afterward write without inverted comma if possible.
  • Line 469-470: Please rewrite this sentence
  • Line 472 (100°C): The authors should space between the number and degree centigrade symbol.
  • Line 478-480: Please rewrite this sentence
  • In this manuscript, the authors wrote many long sentences. The long sentences should be split.
  • This manuscript should improve English from a professional proofread company/Native English speaker.
  • The conclusion should precisely write based on the best findings.
  • References should check clearly. Check the Journal rules and regulations. Before submission, it is mandatory to check the journal reference writing style. Some references wrote the full journal name (Ref 1, ---) and somewhere wrote the short journal name (Ref 2, ---).
  • The authors used 107 references—too many. I think the authors should use recent and relevant ones, skip the old ones.

Round 2

Reviewer 2 Report

 Accept in present form